complexity/statistical physics

urban indicators, maximum entropy, scaling

**Author for correspondence:**
Bernardo Monechi
e-mail: bernardo.monechi@sony.com

# Hamiltonian modelling of macro-economic urban dynamics

Bernardo Monechi[1], Miguel Ibáñez-Berganza[2] and Vittorio Loreto[1,2,3]

[1]Sony Computer Science Laboratories, 6, Rue Amyot, 75005 Paris, France
[2]Physics Department, Sapienza University of Rome, Piazzale Aldo Moro 2, 00185 Rome, Italy
[3]Complexity Science Hub Vienna, Josefstädter Strasse 39, 1080 Vienna, Austria

 BM, 0000-0003-3841-3989; MI-B, 0000-0002-5981-1380

The rapid urbanization makes the understanding of the evolution of urban environments of utmost importance to steer societies towards better futures. Many studies have focused on the emerging properties of cities, leading to the discovery of scaling laws mirroring the dependence of socio-economic indicators on city sizes. However, few efforts have been devoted to the modelling of the dynamical evolution of cities, as reflected through the mutual influence of socio-economic variables. Here, we fill this gap by presenting a maximum entropy generative model for cities written in terms of a few macro-economic variables, whose parameters (the effective Hamiltonian, in a statistical-physical analogy) are inferred from real data through a maximum-likelihood approach. This approach allows for establishing a few results. First, nonlinear dependencies among indicators are needed for an accurate statistical description of the complexity of empirical correlations. Second, the inferred coupling parameters turn out to be quite robust along different years. Third, the quasi time-invariance of the effective Hamiltonian allows guessing the future state of a city based on a previous state. Through the adoption of a longitudinal dataset of macro-economic variables for French towns, we assess a significant forecasting accuracy.

## 1. Introduction

One of the significant challenges humanity is currently facing is accelerated urbanization. According to the UN, some 55 per cent of the global population lives in cities, and this fraction is expected to rise to more than two-thirds by 2050. Different scientific communities accepted the challenge and have started to build a deep understanding of the phenomena related to the urban environment, to develop more sustainable and livable cities.

One of the more interesting recent findings in the field of the Science of Cities is the so-called scaling laws in urban indicators. According to these laws, the population $P$ is the crucial determinant for cities, and other macro-economic features of a city, say $X$, depend on $P$ through a power-law $X \sim P^\beta$ with a feature-dependent exponent $\beta$ [1–5]. Some quantities appear to scale superlinearly with $P$ (i.e. $\beta > 1$), for instance, the GDP or the number of serious crimes, while others depend sublinearly on $P$ (i.e. $\beta < 1$), e.g. the number of infrastructures [6]. These scaling laws appear as a fundamental property of urban environments, naturally emerging from their growth dynamics [7]. Recently, some criticisms have been raised about the concept of scaling in urban systems [8]. On the one hand, it has been shown how it is hard to distinguish $X \sim P^\beta$ from a linear dependency on $P$; on the other hand, the exponent $\beta$ might depend on how one defines city boundaries [9–11].

Despite these issues, scaling laws have profound consequences in the way we think about cities. Albeit cities of different size exhibit very different macro-economic features, when described in terms of rescaled variables, they behave in a size-independent way. Consequently, scaling laws allow for a characterization of cities as abstract, size-independent, entities operating at different scales defined by the population size. Such an intriguing idea has been backed up in time by empirical observations as well as various modelling schemes, trying to grasp the microscopic mechanism responsible for the emergence of scaling. While the identification of the mechanisms behind the emergence of scaling laws is essential to understand the evolution of cities, the current research is still lacking studies aimed at understanding how different indicators influence each other.

To fill this gap, we present here a maximum entropy (ME) generative model for cities written in terms of a few macro-economic variables, whose parameters (the effective Hamiltonian, in a statistical-physical analogy) are inferred from real data through a maximum-likelihood approach.

In our approach, we assume that scaling laws are an intrinsic property of cities. Focusing on indicators related to the job market (e.g. employment rate, number of jobs in the tertiary, etc.), we exploit scaling laws to define population-independent macro-economic indicators, through which we construct the model (we refer to the electronic supplementary material, for further details). The ME inference principle [12] on which our modelling scheme relies has a longstanding history of successful applications in statistical physics [13], biology [14–17], along with other interdisciplinary applications [18,19]. The ME principle guarantees, on rigorous information-theoretical grounds, that the generative model is the most general probability distribution in terms of the considered set of indicators that reproduces only the statistically significant database statistics, under absence of any other assumption or artefact. The model parameters, i.e. the coupling parameters of the Hamiltonian function, are *inferred* following a maximum-likelihood principle, from a longitudinal dataset composed by about 11 000 French 'communes' (the smallest administrative French units ranging from areas of few inhabitants to large metropolis) in 10 different years.

Our new modelling scheme allows to establish three main results. First, thanks to its nonlinear character, the inferred generative model goes beyond the multivariate Gaussian distribution of the indicators and allows us to reproduce the non-trivial empirical correlations among rescaled features accurately. Consequently, the model is not only constrained to reproduce the covariance among pairs of indicators but eventually also the couplings among triplets and quadruplets of indicators. Second, our analysis reveals that the model parameters inferred from distinct year data turn out to be statistically indistinguishable. Third, and more importantly, our modelling scheme features a significant forecasting accuracy of the future state of a city based on a previous state of it. The quasi-stationarity of the coupling parameters just mentioned, suggests the possibility to describe the evolution of urban macro-economic indicators as the solution of a stochastic differential equation of the Langevin type. Though in the literature cities are often described as out-of-equilibrium systems [5,20], we observe that assuming a quasi-equilibrium dynamical evolution of rescaled indicators allows to forecast with high precision their future value in individual cities. To this end, we assume that the vector of indicators obeys the solution of a discretized Langevin equation whose stationary state is given by the previously inferred Hamiltonian. Interestingly, our model can forecast the next-year vector of urban features despite the model parameters having been inferred from single-year empirical data, i.e. using no information regarding the temporal evolution. Indeed, the results are as accurate as the best (maximum likelihood) linear causality inference model that has been trained exploiting the temporal order in the whole time interval.

We believe that this framework may allow for a better understanding of urban environments and their evolution. Unlike other inference models suffering from the black-box problem, our ME approach offers a more precise interpretation of the effective mutual influence among the different macro-economic indicators in a given country or region.

The outline of the paper is as follows. In §2.1, we introduce the data, the relevant observables derived from it and the ME model build using such observables. In §§2.2 and 2.3, we test the stationarity of the model by comparing the parameters inferred in different years, and we derive a discrete model for temporal predictions using the Langevin Equation. In §2.4, we use this model to predict the evolution of individual cities in subsequent years, comparing such predictions with those obtained with a model that explicitly uses the temporal correlations present in the data.

# 2. Results

## 2.1. Correlations of rescaled socio-economic indicators

The data considered in our analysis comes from the INSEE (the French Institut National de la Statistique et des Études Économiques),[1] for French *communes* from 2006 to 2015. We use this data to build $N$ macro-economic indicators representing the job market (jobs in Primary and Secondary Sectors, in the Tertiary and Quaternary Sectors, in Commerce, in Public Administration and services, the Employment rate) and some demographics of each commune (fraction of highly educated people, number of immigrants, average salary per hour).

In electronic supplementary material, S1, we report the code of each INSEE data variable used to build our indicators, as well as the corresponding INSEE dataset used. The indicators used in our analysis are related to demographic or economic aspects of the French population. Nevertheless, the modelling scheme allows for the inclusion of other kinds of indicators, such as crime rates, commercial links between cities, and migrations between communes.

We indicate a generic socio-economic indicator as $X_i^{(\alpha)}$, where $i$ is the index of the indicator, $i = 1, \ldots, N$, and $\alpha$ indicates the commune the indicator refers to. We now define the *rescaled indicators*, $x_i^{(\alpha)}$, as

$$x_i^{\prime(\alpha)} = \log_{10}\left(\frac{X_i^{(\alpha)}}{(X_i^0 P_\alpha^{a_i})}\right), \tag{2.1}$$

where $P_\alpha$ is the population in the commune and $a_i$ is the exponent of the scaling law associated with the $i$-th indicator. $X_i^0$ is the prefactor of the dependence of $X_i$ on the population size, $X_i = X_i^0 P_\alpha^a$. Finally, we divide each indicator by its standard deviation $x_i = x_i'/\sigma(x_i')$. The scaling procedure and the standardization might harden the readability of the variables themselves, as compared to the standard way to present socio-economic indicators. Nevertheless, this procedure makes all the variables living in similar spaces and allows for the comparison of communes of different sizes. For readability's sake, if a rescaled indicator of a specific commune is exactly 0, it means that the actual indicator is precisely the average of all the communes with the same population. Similarly, if the rescaled indicator is 1, then the value of the actual indicator is one standard deviation larger than the average over the communes with the same population.

Recent studies have focused on several aspects of the standardized indicators, $x_i$. In [3], it is shown how they exhibit fast decaying spatial correlations. In [6,20,21], it has been shown how scaling laws by themselves are not sufficient to predict the evolution of cities. In this work, we are interested in building a probabilistic generative model in terms of the vector of rescaled indicators, $\mathbf{x} = (x_i)_{i=1}^N$. We will call $\mathcal{P} : \mathbb{R}^N \to \mathbb{R}$ the probability distribution defining the model, and $< \cdot >_\mathcal{P}$ the expectation value according to it. The generative model is required to reproduce the empirical correlations up to the $m$-th order. To do so, we need to estimate the order of the correlations, $m$, that is relevant and sufficient to describe the data, given the uncertainty associated with the database finiteness. We define the empirical $n$-th order tensor of correlations as

$$C_{i_1, \ldots, i_n}^{(n)} = \langle x_{i_1} \ldots x_{i_n} \rangle_{\text{data}}, \tag{2.2}$$

where $\langle \cdot \rangle_{\text{data}}$ indicates the empirical average over the communes belonging to the database (i.e. over the index $\alpha$). Also, we refer to $n$-th order tensor of cumulants, $\bar{C}^{(n)}$. For each order $n$, we have computed the fraction of elements of the tensors $C^{(n)}$ and $\bar{C}^{(n)}$ that are significantly different from zero given their statistical error. To this end we adopted the bootstrap error, which accounts for the empirical uncertainty induced by the database finiteness (see the electronic supplementary material for details), while the statistical significance refers to a Student $t$-test. The non-significance of the $n$-th order

---

[1] https://www.insee.fr/fr/accueil

cumulants indicates, at least, that they cannot be significantly measured due to the database finiteness (this is to be expected, especially for large $n$). Consequently, they should not be considered as sufficient statistics to be reproduced by $\mathcal{P}$ or, in other words, that $m < n$. Conversely, the presence of significantly non-zero values of the cumulant $\bar{C}^{(n)}$ imply that one should ask the model to reproduce them (i.e. $m \geq n$). If the $n$-th order correlator is non-zero, this does not imply that $m \geq n$, since they could be due explained by lower-order correlations. For example, even for Gaussian data (for which $m = 2$), the fourth-order correlator $C^{(4)}$ is non-zero in general, while the fourth-order cumulants ($\bar{C}^{(4)}_{ijkl} = C^{(4)}_{ijkl} - C^{(2)}_{ij}C^{(2)}_{kl} - C^{(2)}_{ik}C^{(2)}_{jl} - C^{(2)}_{il}C^{(2)}_{jk}$) vanish. We observe that ($n = 1$) all the averages, $C^{(1)}_i$, of the features are consistent 0 with a $p$-value larger than 0.05; ($n = 2$) approximately 91% of two-point correlations, $C^{(2)}_{ij}$, are non-zero ($p < 0.05$); ($n = 3$) approximately 61% of three-point correlations, $C^{(3)}_{ijk}$, are non-zero ($p < 0.05$). We, hence, conclude that $m \geq 3$. ($n = 4$) While approximately 63% of four-point correlations are non-zero ($p < 0.05$), only approximately 6% of the cumulant components $\bar{C}^{(4)}_{ijkl}$ are significantly non-zero ($p < 0.05$). We will consequently consider $m = 3$. In other words, we will consider $C^{(2)}$ and $C^{(3)}$ as sufficient statistics that the model $\mathcal{P}$ is constrained to reproduce.

## 2.2. Inference of the Hamiltonian model for cities

The ME framework leads to an *energy-based* probability distribution (a Maxwell–Boltzmann distribution, in the language of statistical physics), $\mathcal{P}(\mathbf{x}) \propto \exp(-H(\mathbf{x}))$, with an associated Hamiltonian functional $H(\mathbf{x})$ in the space of socio-economic indicators, whose form is determined by the sufficient statistics:

$$H(\mathbf{x}) = \sum_{ij} J^{(2)}_{ij} x_i x_j + \sum_{ijk} J^{(3)}_{ijk} x_i x_j x_k - \sum_i J^{(1)}_i x_i, \tag{2.3}$$

where $J^{(2)}$ and $J^{(3)}$ are the coupling parameters for binary and ternary interactions, respectively. The presence of the term with $J^{(1)}$ is required to compensate the effects of the second term with $J^{(3)}$ and make sure that the averages produced by the model are equal to the empirical ones. The introduction of the $J^{(3)}$ couplings would lead to a non-normalizable distribution $\mathcal{P}(\mathbf{x}) \propto \exp(-H(\mathbf{x}))$ if $\mathbf{x} \in \mathbb{R}^N$. We have consequently reduced the support of the distribution to the hypercube $x_i \in [-L, L]$ with $L = 6$. In other words, each indicator in the distribution support can be at maximum six standard deviations away from its average. The resulting probability distribution after learning the data exhibits a single absolute maximum in its support, near the origin $\mathbf{x} = \mathbf{0}$. Qualitatively, it is a perturbation of the multivariate Gaussian distribution (obtained for null values of the tensor $J^{(3)}$) with respect to which it exhibits larger and asymmetrical tails close to the border of the hypercube (see electronic supplementary material, S4 for a more detailed discussion). The coupling parameters are estimated by the maximum-likelihood method. Such a maximization is performed numerically by deterministic gradient ascent (see electronic supplementary material, S7). The theoretical correlation corresponding to the likelihood gradient at each iteration of the gradient ascent dynamics is estimated by means of Markov chain Monte Carlo (MCMC) sampling. Once the parameters have been inferred, we perform a convergence and consistency check by verifying the extent to which the model reproduces experimental correlations of $n$-th order.

Figure 1 shows the comparison between the empirical $C^{(n)}$ with those produced by the model taking into account the correlations up to the order $n = 5$. The synthetic $C^{(n)}$ have been estimated generating a sample from $\mathcal{P}(\mathbf{x}) \propto \exp(-H(\mathbf{x}))$ (using the Langevin equation below, equation (2.5)), which allows also for an estimation of the standard deviation of each component of $C^{(n)}$ (see electronic supplementary material, S6). We can use this error, together with the bootstrapped error of the empirical $C^{(n)}$, to perform a $t$-test of consistency. The percentage of non-compatible components for each $C^{(n)}$ is less than 5%, the few discrepancies typically occurring for points with large estimation errors, especially for $n = 4$ and $n = 5$. This result validates the numerical gradient ascent procedure. Furthermore, the model consistently reproduces correlators at orders $n = 4$ and $n = 5$, which were not supposed to be reproduced by construction. In conclusion, these results *a posteriori* justify and validate our ME approach and the sufficient statistics used, i.e. using cumulants up to $m = 3$.

In order to further validate the above statement, we compared the performances of the model described by equation (2.3) with a simpler Gaussian one in which we removed all terms except the $J^{(2)}$ one, and we let the variables $x_i$ to be defined in the whole space $\mathbb{R}^N$:

$$H_G(\mathbf{x}) = \sum_{ij} J^{(2)}_{ij} x_i x_j. \tag{2.4}$$

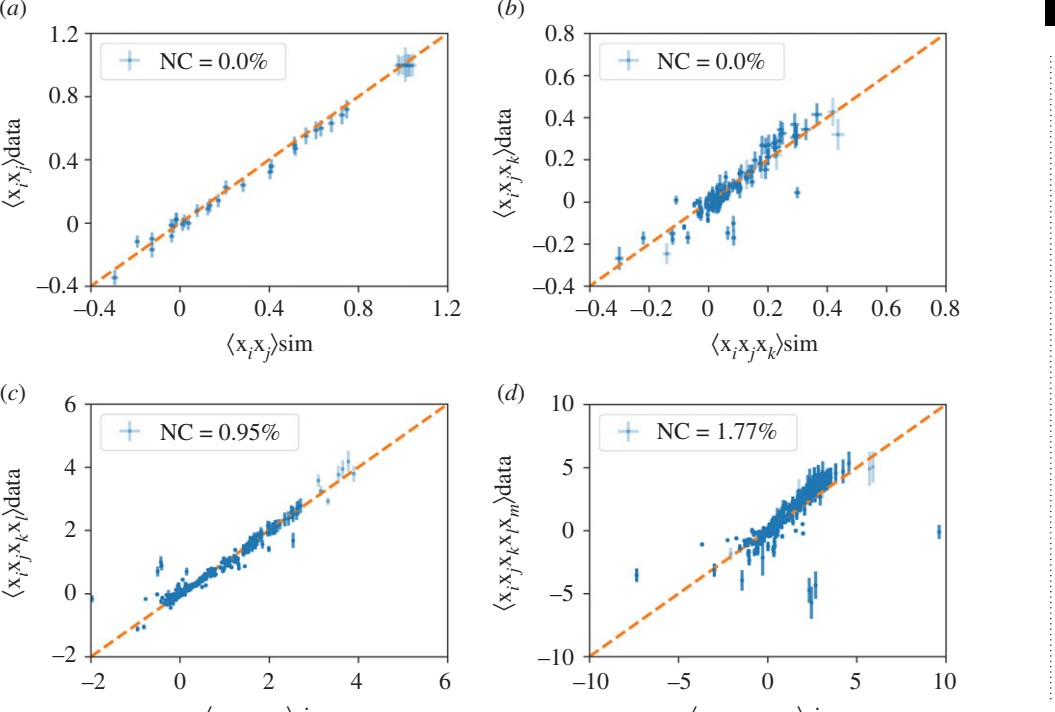

**Figure 1.** Comparisons between the correlators of order 2 (*a*), 3 (*b*), 4 (*c*) and 5 (*d*) obtained with the empirical data of year 2012 (*y*-axis) and the Hamiltonian model (equation (2.3)) (*x*-axis). We report the percentage of components of each correlation which is not compatible with the data via a *t*-test with *p*-value 0.05 (see electronic supplementary material, S6).

The experiment aims to evaluate which model better grasps the interplay between different socio-economic indicators. To illustrate the results, we show, without loss of generality, the dependence of one indicator as a function of the other two. Figure 2 reports the comparison of the models' predictions with the empirical data. In particular, we show the dependence of one rescaled indicator (i.e. *Jobs in Quaternary Sector* in panel (a) and *Jobs in Primary and Secondary Sectors* in panel (b)) on other two rescaled indicators, namely *Jobs in Public Administration* (*x*-axis) and *Fraction of highly educated people* (*y*-axis).

It is evident that the introduction of the nonlinear term $J^{(3)}$ increases the model ability to predict, and it is key to capture the nonlinear effects present in the data. Given a generative model $\mathcal{P}(\mathbf{x})$, it is possible to study this dependency by sampling from the conditional probability $\mathbf{x}$ ($x_i | x_j, x_k$) for several values of ($x_j, x_k$), being $x_i$ the considered dependent variable. The first column of figure 2, shows this dependency in the case of a model inferred without the terms $J^{(1)}$ and $J^{(3)}$, a simple Gaussian model leading to linear dependencies among all the variables. The prediction of this model is not in agreement with the data, shown in the last column of figure 2. Instead, the inclusion of $J^{(1)}$ and $J^{(3)}$ (central panels) leads to more adherence to the empirical data. Considering, for example, the upper right corner of all the panels, this represents communes with a large number of jobs in public administration and a large number of residents with high education. In this area, both the linear and nonlinear models agree with the data, predicting a large number of Jobs in Quaternary (the rescaled indicator is around 1) and an average number of Jobs in the Primary and Secondary sectors (the rescaled indicator is around 0). Moving instead to the lower-left corner of each panel, this represents communes with very few Jobs in Public Administration and residents with high-education. In this area, the linear model would predict an average number of Jobs in the Primary and Secondary and very few Jobs in Quaternary (the rescaled indicator is −3). However, the data show that in this corner, there is a large number of jobs in both sectors. This fact can be easily explained by the presence in our dataset of mainly industrial areas, poorly served by public administrations and with a scarcely educated population in which workers of every skill commute to work. This behaviour is correctly predicted by the model in equation (2.3). Other examples, similar to that in figure 2, can be found in electronic supplementary material, S5, fully confirming this picture.

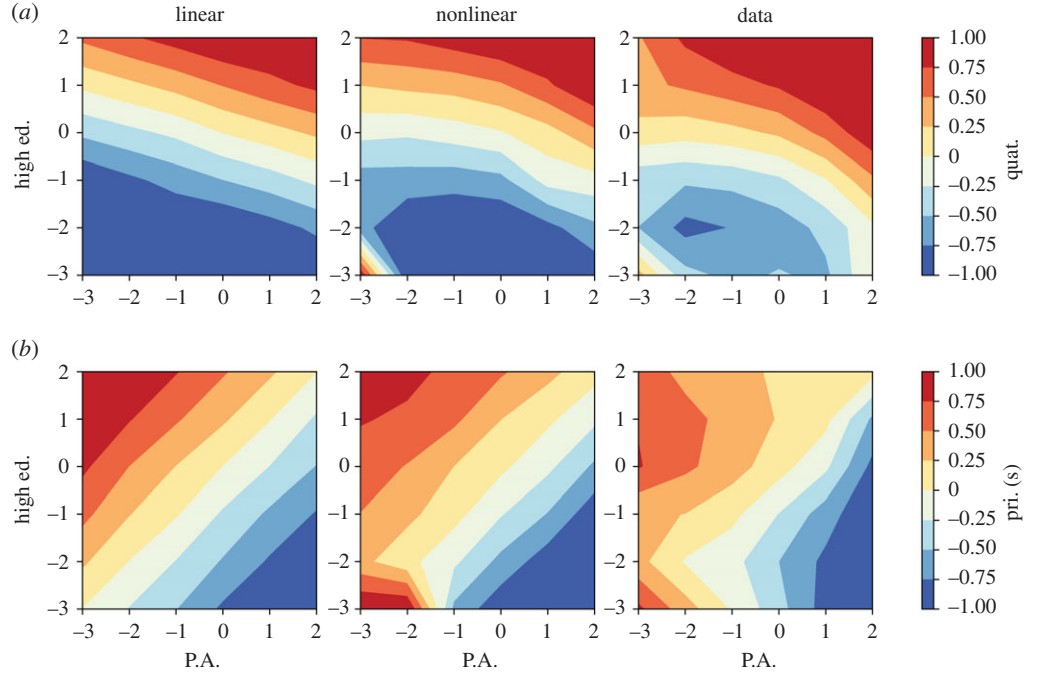

**Figure 2.** Rescaled indicators for *Jobs in Quaternary Sector* (*a*) and *Jobs in Primary and Secondary Sectors* (*b*) as functions of the rescaled indicators for *Jobs in Public Administration* (*x*-axis) and *Fraction of highly educated people* (*y*-axis). Areas in red (blue) represent communes with a large (small) value of the rescaled indicator used as dependent variable. The first column (with the label *linear*) reports the results obtained with the Hamiltonian model without the terms $J^{(1)}$ and $J^{(3)}$. The second column (with the label *nonlinear*) reports the results obtained with the complete model of equation (2.3). The last column reports the results obtained by binning the points for the communes in the year (2012).

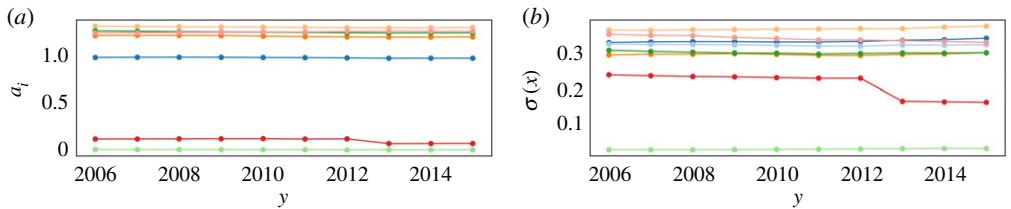

**Figure 3.** (*a*) Scaling exponents $a_i$ for the socio-economic indicators *i* as a function of time. (*b*) $\sigma(x_i)$ for the socio-economic indicators *i* as a function of time. Each indicator is represented by a different colour.

## 2.3. Stationarity of the model

A very important question to ask is whether the modelling scheme proposed in the section above, and synthesized by equation (2.3), is robust with respect to the empirical data gathered in different years. In other words, are the values of the interaction parameters $J^{(n)}$ stable when inferred from different years' data? The answer is yes as we illustrate below.

First, we observe from figure 3 that most of the exponents of the scaling law, $a_i$, used to define the rescaled indicators through equation (2.1) are constant in time, and it is so also for the standard deviations $\sigma(x_i)$. Small deviations are only seen for two indicators.

Having checked the stationarity of the quantities used to define the variables $x_i$, we have addressed the stationarity of the inferred interaction parameters $J^{(n)}$. This is done through a significance *t*-test (see electronic supplementary material, S6) of the compatibility between $J^{(1)}(y_1)$, $J^{(2)}(y_1)$, $J^{(3)}(y_1)$ inferred in a certain year $y_1$, and $J^{(1)}(y_2)$, $J^{(2)}(y_2)$, $J^{(3)}(y_2)$ in year $y_2$. Indeed, the parameters are statistically compatible across different years (*p*-value < 0.05, see figure 4; electronic supplementary material, S6 for the comparisons of $J^{(1)}$ and $J^{(3)}$). Hence, despite the moderate variation in the scaling exponents and the standard deviations of the indicators, the model is stationary from one year to another. This result does not imply that the indicators of a single commune, $\mathbf{x}^{(\alpha)}$, are not evolving in time. The values of

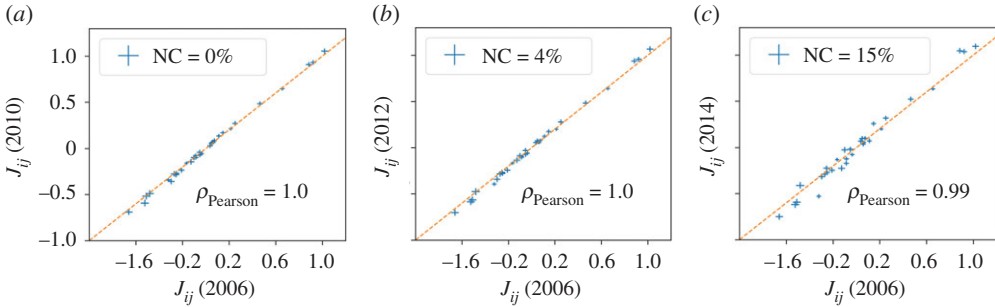

**Figure 4.** Comparisons between the $J^{(2)}$ parameters of different years (2006–2010 in (*a*), 2006–2012 in (*b*), 2006–2014 in (*c*)). The dotted line represents the relation of equality (i.e. the diagonal of the first half-plane in the Cartesian space. The percentage of non-compatible (NC) reported refers to the percentage of components that cannot be considered as equal with a *t*-test and a threshold *p*-value of 0.05.

$\mathbf{x}^{(\alpha)}$ are, in fact, *non-stationary* from one year to another. Instead, the correlations among several indicators, $C^{(n)}$, stay constant. This remarkable result paves the way for a description of the evolution of socio-economic indicators in terms either of *equilibrium models*, in a statistical-physical sense, or out-of-equilibrium stationary models [22].

## 2.4. Forecasting the time evolution of the socio-economic indicators

In this section, we test the forecasting capacity of our Hamiltonian modelling scheme, i.e. the ability to predict future values of the rescaled indicators starting from a given starting condition. To this end, we consider the Langevin equation for the stochastic temporal evolution of a vector field [23]. This provides a simple model for the continuous-time dynamics of vector $\mathbf{x}$, whose stationary distribution is our generative model, $\mathcal{P}(\mathbf{x}) \propto \exp(-H(\mathbf{x}))$:

$$\frac{\mathrm{d}\mathbf{x}(t')}{\mathrm{d}t'}(t) = -\nabla H(\mathbf{x}(t)) + \boldsymbol{\eta}(t), \tag{2.5}$$

where $\boldsymbol{\eta}(t)$ is $N$-dimensional vector of independent random variables with vanishing average, extracted from a probability distribution $h$, satisfying $\langle \eta_i(t)\eta_j(t')\rangle_h = \delta(t-t')\delta_{ij}$. This is a strong assumption, that implies not only the stationarity of the distribution of $\mathbf{x}$ in the large $t$-limit but also *thermal equilibrium* (or, roughly speaking, absence of probability currents) [22]. Nevertheless, this assumption might still be useful to make predictions about the trajectories of individual cities. The distribution of $\boldsymbol{\eta}$s, $h(\boldsymbol{\eta})$ can be chosen arbitrarily. We use a Laplacian noise, i.e. $h(\boldsymbol{\eta}) \propto \exp(-|\boldsymbol{\eta}|/2)$ (see electronic supplementary material, S7). In equation (2.5), the time is a continuous variable, whose physical interpretation is not straightforward for our model, since our data is defined in discrete time. We will indicate different years with a specific intrinsic time $t_y$ so that the consecutive year time is $t_{y+1}$. Approximating the derivative by a finite difference, $dt = t_{y+1} - t_y$, we can derive the probability of observing the feature vector $\mathbf{x}(t_{y+1})$ for a certain city after having observed the values of the previous year $\mathbf{x}(t_y)$ as

$$p_{dt}(\mathbf{x}(t_y + dt)|\mathbf{x}(t_y)) = \prod_{j=1}^{N} \sqrt{\frac{1}{2dt}} \exp\left(-\frac{\sqrt{2}|x_j(t_y + dt) - x_j(t_y) - f_j(\mathbf{x}(t_y))\,dt|}{\sqrt{dt}}\right), \tag{2.6}$$

where $f_j(\mathbf{x}(t_y)) = -\frac{\partial H}{\partial x_j}(\mathbf{x}(t_y))$. Assuming that our system is governed by equation (2.5), we can use maximum likelihood to estimate the value of $dt$ that best reproduces the transitions between subsequent years (see electronic supplementary material, S7). According to the discrete-time Langevin model, the variation of a feature vector from one year to the next, $\boldsymbol{\Delta}(t_y) = \mathbf{x}(t_{y+1}) - \mathbf{x}(t_y)$ should be proportional to minus the gradient of the Hamiltonian $H$ plus some Laplacian noise. Hence, such variation should be on average parallel and proportional to $-\nabla H(\mathbf{x}_y)$ (where $\mathbf{x}_y = \mathbf{x}(t_y)$). To check this hypothesis, we compare the angle $\omega_{\mathrm{data}}$ between two consecutive variations of the feature vector $\boldsymbol{\Delta}(t_y)$ and $\boldsymbol{\Delta}(t_{y+1})$, with the angle $\omega_{\mathrm{model}}$ between $-\nabla H(\mathbf{x}(t_y))$ and $\boldsymbol{\Delta}(t_y)$. Figure 5*a* shows this comparison. We can interpret $\omega_{\mathrm{data}}$ as the angle between two consecutive velocities of the system, while $\omega_{\mathrm{model}}$ is the angle between the velocity at time $t_y$ and the predicted velocity at time $t_{y+1}$ (i.e. $-\nabla H(\mathbf{x}(t_{y+1}))dt$). The agreement between the angles indicates that the rotation of the velocity at different times is compatible with that predicted by the Langevin dynamics. The remarkable agreement between the data and the synthetic sample

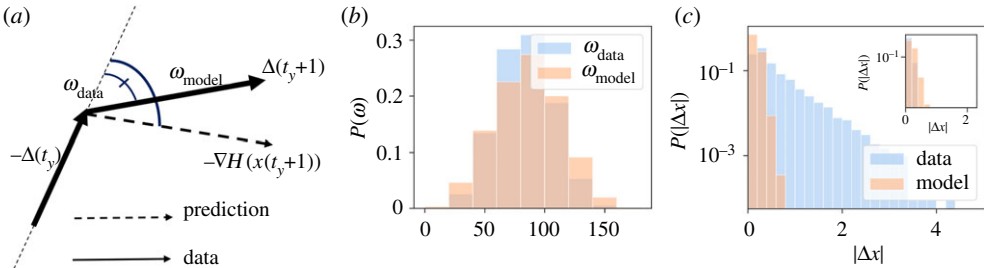

**Figure 5.** (a) Graphical representation of the angles $\omega_{\text{data}}$ (the smaller angle in (a)) and $\omega_{\text{model}}$ (the larger angle in (a)), identified respectively by the variation $\Delta(t_{y+1})$ at time $t_y$ and the predicted variation of the model at the same time $-\nabla H(\mathbf{x}(t_y))$, and by the two subsequent variations $\Delta(t_y)$ and $\Delta(t_{y+1})$. (b) Comparison between the angle $\omega_{\text{data}}$ between the velocity of the system at consecutive times and the angle $\omega_{\text{model}}$ between the velocity of the system and the velocity predicted by equation (2.5). (c) Variations $\Delta x_i(t_y)$ for all the components $i$ of the feature vector in the same two cases. In the inset, we show the same comparison excluding communes with a population larger than $10^4$.

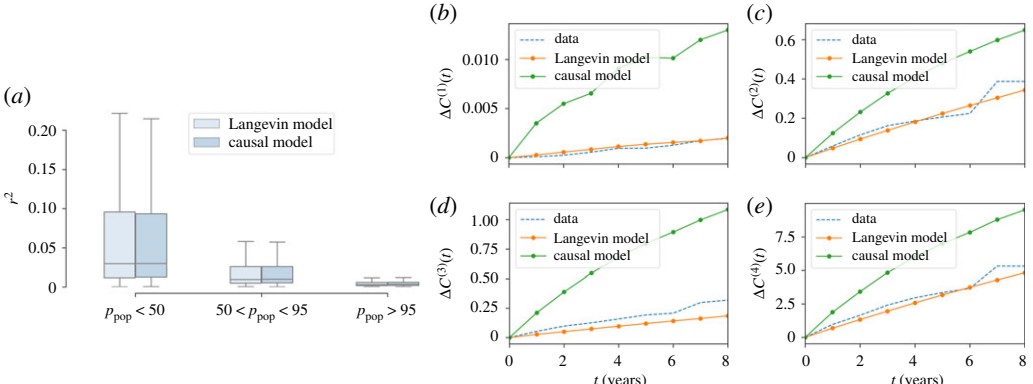

**Figure 6.** (a) Distribution of $r^2$ computed according to equation (2.7) and with a causal inference model, inferred using temporal information explicitly. The values of $r^2$ have been divided according to the percentile of the commune population distribution. (c,b,d, e) Evolution of the distance between macroscopic observables computed in different years with respect to those computed in 2012 ($t = 0$ on the x-axis).

justifies equation (2.5) as a model of the evolution of the urban indicators. A comparison between the modules of each $\Delta(t_y)$ obtained with data and with simulations is shown in figure 5c. In this case, the agreement is less strong, since the real data distribution is broader for extreme values. However, if we restrict the comparison only to large communes, with a population $P > 10^4$, the agreement increases (inset of figure 5c), suggesting that the discrete-time Langevin approach, equation (2.5), is, at least, a good model for the evolution of large cities. This fact emerges also from a further analysis of the model dynamical forecasting accuracy. We have performed a statistical test to evaluate the accuracy of the model prediction for $\mathbf{x}(t_{y+1})$ from the real data $\mathbf{x}(t_y)$, according to equation (2.6). For every city and every year $y$, we consider the quantity

$$r_y^2 = \frac{1}{N} \sum_{j=1}^{N} |x_j(t_{y+1}) - x_j(t_y) - f_j(t_y) \, dt|^2, \qquad (2.7)$$

i.e. the average square residual between the actual feature vector at time $t_{y+1}$, $\mathbf{x}(t_{y+1})$, and the average model prediction, i.e. $\mathbf{x}(t_y) - \mathbf{f}(t_y) dt$. Figure 6a shows the distribution of the $r_y^2$ variables after dividing the sample according to the percentile of population of each city. The model performs better as the size of the city increases, in agreement with the results in figure 5. In order to have an accuracy baseline, figure 5 shows the same $r_y^2$ divided according to the population for a causal inference (CI) model [13] in which the information about temporal correlations of the indicators at consecutive and non-consecutive years are explicitly inferred by maximum likelihood. Both distributions are statistically compatible. It is strikingly surprising that the accuracy of both models are equivalent,

since the discrete-time Langevin model *has been inferred from the single-year data*, hence *neglecting the database information regarding the time evolution*. Yet, our model forecasts rather accurately. In the CI, the information regarding the temporal evolution is *inferred* from the data. In our discrete-time Langevin model it is, instead, *postulated* through equation (2.6), and does not need to be inferred. This, however, comes at the cost of introducing a non-stationary term in the model, absent in the discrete-time Langevin dynamics. We can show this computing the observables $C^{(n)}(y)$ with $n = 1, 2, 3, 4$ from equation (2.2) for each year $y$ of our data. For each $n$, we define $\Delta C^{(n)}(y) = |C^{(n)}(y) - C^{(n)}(0)|_2$, where $|\cdot|_2$ is the Frobenius norm. This quantity indicates for each year, how much the observable $C^{(n)}(y)$ has shifted from its initial value $C^{(n)}(0)$. Thus, we produce two synthetic samples obtained by making each commune in the first year of our data evolve according to the Langevin equation and to the CI model. By computing the observables $C^{(n)}(y)$ for each time step of the two synthetic samples, we can compute the corresponding value of $d^{(n)}(y)_{\text{Langevin}}$ and $d^{(n)}(y)_{\text{Causal}}$. Figure 6b,c,d,e shows $d^{(n)}(y)$ as a function of $y$ for the data, the Langevin model and the causal model. We see that the data shows a small shift in the observables and the Langevin model is always more coherent the causal model in reproducing it.

## 3. Conclusion

In relatively recent times, the phenomenon of urbanization is proceeding at an unprecedented pace. Nowadays, urban environments represent the pumping heart of modern-day life with all its diverse aspects affecting progress and innovation. Despite the importance of the phenomenon, little is known about the critical determinants of cities and their evolution. It has been observed that socio-economic indicators related to urban environments follow scaling laws with the city size. Those regularities helped to formulate hypotheses about the deep meaning of the observed self-similarity as well as the mechanisms for the emergence of these laws. One of the big successes of scaling theory applied to city science is the possibility it opens to define rescaled socio-economic indicators, which, in their turn, allow for comparing different cities at different population scales. Despite these successes, scaling theory represents an *a posteriori* description of cities, and little can be said *a priori* about the dynamical evolution of these relevant entities and their constituents. However, a modelling framework is still lacking in the science of cities that, through a careful description of the interactions and the couplings among the diverse aspects of the urban fabric, could allow us to assess the status of cities and create validated scenarios of future evolution.

With this paper, we made a step forward in this direction by proposing a first ME generative model of towns based on careful observation of modern cities as witnessed by data of French 'communes' in the period from 2006 to 2015. This generative model defines a Hamiltonian written in terms of a vector of socio-economic indicators whose coupling parameters are inferred through an unsupervised maximum-likelihood approach. The Hamiltonian defines a probabilistic model that takes into account nonlinear effective interactions up to the order $m = 3$ (i.e. it takes into account couplings of two and three socio-economic indicators). In this way, our approach goes beyond a principal component analysis, and it allows reproducing the nonlinear correlations observed in the empirical data up to the fifth order. We show that the inclusion of these couplings is necessary to correctly describe the data, which exhibit in many cases behaviours that are far from linear. The whole approach allows for projecting cities in a high-dimensional landscape (defined in terms of the socio-economic features) where each existing town sits in a specific spot, and its dynamics occurs along the manifold defined by the Hamiltonian model.

Interestingly, the inferred model is quite robust, and the different coupling parameters turnout to be invariant over several years. Along with the stationarity of the scaling laws, this result suggests that the statistical laws governing the socio-economic indicators can be approximately considered as constants in time. If we adopt the terminology of stochastic processes, the stationarity of the inferred model allows for a description of the dynamics of an individual city in terms of stationary out-of-equilibrium or a quasi-equilibrium model. Following the latter and most straightforward approach, we have proposed a dynamical model based on the Langevin equation, consistent with the previously inferred Boltzmann probability distribution (defined by an effective Hamiltonian function), and assessed its predictive power. The forecasting accuracy of the postulated discretized Langevin equation for the dynamical evolution of single cities turns out to be compatible with that of the best (maximum likelihood) linear model of CI, in which the dynamical evolution has not been postulated but inferred from the data.

Our results pave the way for a novel and precise, yet interpretable, predictive modelling of urban environments from a macro-economic point of view. Our framework is also suited to be applied to a CI of the effects of shocks, stress conditions or exogenous events, and to model the recovery of cities

after them. This whole framework could help to forecast the decline or growth of towns and shed light on the causes of such behaviours. For example, a variation in the model parameters could model the effects of changing the national and international scenario on the urban system, as well as the impact of policies in the job market.

Data accessibility. The data analysed in this work is freely available on Figshare at the following link https://figshare.com/articles/Dataset_for_Hamiltonian_Modelling_of_Macro-Economic_Urban_Dynamics_/12196152 (doi:10.6084/m9.figshare.12196152.v1). The uploaded file contains all the socio-economic indicators, the scaling-exponents and the populations of the communes from 2005 and 2016.

Authors' contributions. All authors conceived and designed the research work; B.M. and M.I.-B. ran the simulations and analysed the data; all authors wrote and reviewed the article; all authors gave final approval for publication.

Competing interests. The authors declare no competing interests related to this work.

Funding. There is no specific funding to declare related to this work.

Acknowledgements. We thank Giovanni Cerulli and Andrea Gabrielli for their comments.

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
