## [Reviewer comments · Royal Society Open Science]

Review History

RSOS-200667.R0 (Original submission)

Review form: Reviewer 1

Is the manuscript scientifically sound in its present form?

Yes

Are the interpretations and conclusions justified by the results?

Yes

Is the language acceptable?

Yes

Do you have any ethical concerns with this paper?

No

Have you any concerns about statistical analyses in this paper?

No

Recommendation?

Accept as is

Comments to the Author(s)

In the introduction, the authors comprehensively embrace the main advantages of their proposal, which they define in more detail in the continuation of the article. I see an important contribution in the forecasting the time evolution of socio-economic indicators (Eq. 5) by using a strong assumption including stationarity of the distribution of x in the large t -limit, and the thermal equilibrium as well. The model works well for larger cities. Additional materials, in particular in the section S5, underline the results in the main text.

Review form: Reviewer 2

Is the manuscript scientifically sound in its present form?

No

Are the interpretations and conclusions justified by the results?

No

Is the language acceptable?

Yes

Do you have any ethical concerns with this paper?

No

Have you any concerns about statistical analyses in this paper?

No

Recommendation?

Major revision is needed (please make suggestions in comments)

Comments to the Author(s)

This is a very short report replacing the one I initially wrote on the electronic system but that got lost. My major point is that the 3rd degree Hamiltonian proposed by the authors does NOT lead to a normalised probability distribution. There are **always** directions along which the probability will grow unbounded.

This is not necessarily a problem if some well motivated regularisation is introduced. Think of the 1d case $V(x) = x^2/2 - a/3 x^3$. When $a > 0$ this diverges negatively when x goes to $+\infty$. But there is a minimum of $V(x)$ at $x=0$ around which the probability is "metastable" provided x cannot go beyond $x=a$.

So the question is: what implicit regularisation did the authors make in their analysis to get their results? Does it make sense? This is, I think, a crucial question that will determine whether the work is publishable or needs to be significantly revised.

Decision letter (RSOS-200667.R0)

Dear Dr Monechi,

The editors assigned to your paper ("Hamiltonian Modelling of Macro-Economic Urban Dynamics") have now received comments from reviewers. We would like you to revise your paper in accordance with the referee and Associate Editor suggestions which can be found below (not including confidential reports to the Editor). Please note this decision does not guarantee eventual acceptance.

Please submit a copy of your revised paper before 28-Aug-2020. Please note that the revision deadline will expire at 00.00am on this date. If we do not hear from you within this time then it will be assumed that the paper has been withdrawn. In exceptional circumstances, extensions may be possible if agreed with the Editorial Office in advance. We do not allow multiple rounds of revision so we urge you to make every effort to fully address all of the comments at this stage. If deemed necessary by the Editors, your manuscript will be sent back to one or more of the original reviewers for assessment. If the original reviewers are not available, we may invite new reviewers.

- Data accessibility

If you wish to submit your supporting data or code to Dryad (<http://datadryad.org/>), or modify your current submission to dryad, please use the following link:
<http://datadryad.org/submit?journalID=RSOS&manu=RSOS-200667>

- Competing interests

- Authors' contributions

- Acknowledgements

- Funding statement

on behalf of Dr Robert MacKay (Associate Editor) and Mark Chaplain (Subject Editor)
openscience@royalsociety.org

Associate Editor's comments (Dr Robert MacKay):

Associate Editor: 1

Comments to the Author:

The paper looks interesting but the point raised by reviewer 2 must be addressed before the paper can be considered further.

Comments to Author:

Reviewers' Comments to Author:

Reviewer: 1

Comments to the Author(s)

In the introduction, the authors comprehensively embrace the main advantages of their proposal, which they define in more detail in the continuation of the article. I see an important contribution in the forecasting the time evolution of socio-economic indicators (Eq. 5) by using a strong assumption including stationarity of the distribution of x in the large t -limit, and the thermal equilibrium as well. The model works well for larger cities. Additional materials, in particular in the section S5, underline the results in the main text.

Reviewer: 2

Comments to the Author(s)

This is a very short report replacing the one I initially wrote on the electronic system but that got lost. My major point is that the 3rd degree Hamiltonian proposed by the authors does NOT lead to a normalised probability distribution. There are *always* directions along which the probability will grow unbounded.

This is not necessarily a problem if some well motivated regularisation is introduced. Think of the 1d case $V(x) = x^2/2 - a/3 x^3$. When $a > 0$ this diverges negatively when x goes to + infinity. But there is a minimum of $V(x)$ at $x=0$ around which the probability is "metastable" provided x cannot go beyond $x=a$.

So the question is: what implicit regularisation did the authors make in their analysis to get their results? Does it make sense? This is, I think, a crucial question that will determine whether the work is publishable or needs to be significantly revised.

Author's Response to Decision Letter for (RSOS-200667.R0)

See Appendix A.

Decision letter (RSOS-200667.R1)

Dear Dr Monechi,

It is a pleasure to accept your manuscript entitled "Hamiltonian Modelling of Macro-Economic Urban Dynamics" in its current form for publication in Royal Society Open Science. The comments of the reviewer(s) who reviewed your manuscript are included at the foot of this letter.

on behalf of Dr Robert MacKay (Associate Editor) and Mark Chaplain (Subject Editor)
openscience@royalsociety.org

Associate Editor Comments to Author (Dr Robert MacKay):

Comments to the Author:

You have dealt adequately (in my opinion) with the comments of the reviewers, so I now recommend the paper for publication.

Appendix A

Response to Reviewers for

Hamiltonian Modelling of Macro-Economic Urban Dynamics

by Bernardo Monechi, Miguel Ibáñez-Berganza Vittorio Loreto

Dear Editors,

We wish to thank you for handling the submission of our paper *Hamiltonian Modelling of Macro-Economic Urban Dynamics* and seeking expert reports about it. Thank you also for the appreciation you and the Reviewers had for our work. We have addressed all the points raised by the anonymous Reviewers, and their comments have helped us to improve the quality of the manuscript. In particular, we added some clarifications to the issue raised by Reviewer 2 about the regularisation of the model. It is worth noticing that we regularised the model by constraining the variables into a hypercube centred on the origin of the space. This regularisation prevents the system from diverging, constraining it around a local maximum. The details of the regularisation were present in the previous version and discussed in Section S4 of SI. However, we did not highlight them properly in the main text, and the explanation in S4 was rather insufficient. We hope to have now provided an exhaustive and convincing explanation, and that in its present form our manuscript can be published in Royal Society Open Science. We have highlighted the differences between the old and new documents in red in the latest versions for the main text and the SI.

Below, we report our detailed answers to the Reviewers.

Response to Reviewer 1

In the introduction, the authors comprehensively embrace the main advantages of their proposal, which they define in more detail in the continuation of the article. I see an important contribution in the forecasting the time evolution of socio-economic indicators (Eq. 5) by using a strong assumption including stationarity of the distribution of x in the large t -limit, and the thermal equilibrium as well. The model works well for larger cities. Additional materials, in particular in the section S5, underline the results in the main text.

We thank the Reviewer for her/his kind comments and her/his appreciation for our work. We hope to be able to extend our findings in future works and to explore in more details the consequences of the stationarity assumption. We have made some changes both to Section 2(b) of the main text and to the section S4 of SI that the Reviewer may find in the manuscript with tracked changes. The details introduced concern some clarifications required by the second Reviewer about the regularisation of the model that, albeit was discussed in the previous version of the manuscript, would probably have deserved a more explicit and exhaustive discussion in the main text of the article. Such modifications concern

a technical aspect regarding the regularisation of the probability function for large values of its variables. They do not regard the article's results in any respect. Our conclusions remain unchanged.

Response to Reviewer 2

*This is a very short report replacing the one I initially wrote on the electronic system but that got lost. My major point is that the 3rd degree Hamiltonian proposed by the authors does NOT lead to a normalised probability distribution. There are *always* directions along which the probability will grow unbounded.*

This is not necessarily a problem if some well motivated regularisation is introduced. Think of the 1d case $V(x) = x^2/2 - a/3x^3$. When $a > 0$ this diverges negatively when x goes to $+$ infinity. But there is a minimum of $V(x)$ at $x = 0$ around which the probability is "metastable" provided x cannot go beyond $x = a$.

So the question is: what implicit regularisation did the authors make in their analysis to get their results? Does it make sense? This is, I think, a crucial question that will determine whether the work is publishable or needs to be significantly revised.

We thank the Reviewer for this comment that highlighted this flaw in the manuscript. We were aware of the fact that the introduction of 3-points interaction would lead to an unnormalised probability distribution. We tackled such issue by performing a regularisation procedure consisting in the reduction of the likelihood support. We briefly discussed this regularisation in Section S4 of the SI. We agree with the Reviewer that, however, this critical technical point deserved a more exhaustive and clear explanation, also in the main text, as we now included (see below).

To prevent the model from diverging, we restrict the support of the probability distribution is to a hypercube of length $2L$ centred on the origin, with $L = 6$. In other words, in the model, each indicator could not fluctuate more than 6 standard deviations from its average value (that we set to 0). The hypercube prevents the dynamics of the model from diverging, leaving the local minima outside. The choice of L is quite crucial since a small value would exclude parts of the space populated by real data, while values too large will allow these minima within the hypercube. The inclusion of minima within the hypercube will, in turn, make the training of the model a more difficult task, since the dynamics could get stuck on the borders during the training.

The final result is a model with probability peaked around the origin, and with tails that are broader than the Gaussian case and asymmetric. These deformations are more adherent to the empirical data as we show in SI Section S4. As a side note, we are aware that other choices are possible, for example, the inclusion of other terms in the Hamiltonian (e.g., 4-points interactions) that might work as regularisers. While other choices could be better in terms of prediction accuracy, they are probably more complex to study and train. We

would instead leave the issue of model selection for upcoming works.

In the main text we added in Section 2(b):

The introduction of the $J^{(3)}$ couplings would lead to a non-normalisable distribution $\mathcal{P}(\mathbf{x}) \propto \exp(-H(\mathbf{x}))$ if $\mathbf{x} \in \mathbb{R}^N$. We have consequently reduced the support of the distribution to the hypercube $x_i \in [-L, L]$ with $L = 6$. In other words, each indicator in the distribution support can be at maximum 6 standard deviations away from its average. The resulting probability distribution after learning the data exhibits a single absolute maximum in its support, near the origin $\mathbf{x} = 0$. Qualitatively, it is a perturbation of the multivariate Gaussian distribution (obtained for null values of the tensor $J^{(3)}$) with respect to which it exhibits larger and asymmetrical tails close to the border of the hypercube (see SI section S4 for a more detailed discussion).

In Section S4 of the SI, we explain more in detail this point:

The introduction of the term $J^{(3)}$ in the Hamiltonian is such that the distribution $\mathcal{P}(x)$ cannot be normalised if its support is \mathbb{R}^N . In other words, there will be directions in \mathbb{R}^N that will make the distribution grow indefinitely. However, there might be values of the coupling parameters that will allow for some local maxima that will constrain the dynamics of the system for a finite time, before it diverges for $t \rightarrow \infty$. To prevent this behaviour, we can bound the system around these maxima redefining the support of the probability $\mathcal{P}(\mathbf{x})$ as $I = [-L, L]^N$, i.e., a hypercube centred on the origin. The choice of the value L influences the model training and efficiency in a non-trivial way. If L is too small, some parts of the space that are populated by the empirical data could be excluded. For sufficiently large values of L , the function may develop, during training, a global maximum in I different from the convex, perturbed Gaussian maximum near the origin. Fig. S4 shows the percentage of data points within the hypercube as a function of L . We see that the first value that almost all the sample if $L > 5$, hence we choose $L = 6$, i.e. 6 standard deviations of the sample.

Then we explain the approach used for training, in particular, the need to approximate the gradients and the cost function since the model in the hypercube is not solvable:

To circumvent this problem, we will use an approach widely used for training Energy Based models in Machine Learning [2, 3]. We can write the log-likelihood of our model as:

$$\mathcal{L} = -\langle H(\mathbf{x}) \rangle_{data} - \log Z, \quad (1)$$

Where $\langle \cdot \rangle_{data}$ indicates the average on the sample data. Taking the

gradient of the above expression we find

$$\nabla\mathcal{L} = -\langle\nabla H(\mathbf{x})\rangle_{data} + \langle\nabla H(x)\rangle_{\mathcal{P}}, \quad (2)$$

where $\langle\cdot\rangle_{\mathcal{P}}$ is the average for the model. This average can be approximated at each training step by averaging over a sample obtained with numerical simulations (e.g., by iterating the equation (18)). If we use this approximation we can see from equation (12) that maximising the log-likelihood is equivalent to optimise the cost function:

$$\mathcal{C} = -\langle H(x)\rangle_{data} + \langle H(x)\rangle_{\mathcal{P}}, \quad (3)$$

that we can use to monitor the development of the training.

Finally, we show how the resulting probability distribution differs from the Gaussian case:

Fig. S3 we show the cost function curves for all the train and test data for four different years. After a maximum, the cost function decreases, approaching monotonically zero for large values of the number of iterations. The curves for the training and test sets are indistinguishable for some years, or present non-significant differences. We conclude that there are not overfitting issues: the model generalises well to non-observed data.

We can study how the probability distribution (10) defined in the hypercube differs from a Multivariate Gaussian distribution (9). We will assess the impact of the tensor $J^{(3)}$ in the distribution through a comparison with the Gaussian model in equation (9), in terms of the principal components (PCs), or the projections of the physical variables \mathbf{x} on the eigenvectors of the $J^{(2)}$ matrix. If we do so, the model becomes a set of non-interacting Gaussian models. **Fig. S4** shows the comparison among the linear model (a collection of independent normal distributions over the PCs), the non-linear model and the empirical data, as a function of the first PCs. We can see that on each PC the model (9) predicts a Gaussian distribution centred in 0 (orange line), qualitatively similar to the empirical distribution (blue bars). Doing the same for the model in equation (10) gives a slightly different result. We can see in **Fig. S4** that the introduction of a bounded dominion allows the model to reproduce the empirical bell-shaped distribution (green line). However, the introduction of $J^{(3)}$ modifies the shape distribution tails, which are now more adherent to the empirical one.

Yours sincerely,

Bernardo Monechi

Sony Computer Science Laboratories, 6, Rue Amyot, 75005, Paris, France
bernardo.monechi@sony.com

Miguel Ibáñez-Berganza

Sapienza University of Rome, Physics Department, Piazzale Aldo Moro 2, 00185,
Rome, Italy

Vittorio Loreto

Sony Computer Science Laboratories, 6, Rue Amyot, 75005, Paris, France Sapienza
University of Rome, Physics Department, Piazzale Aldo Moro 2, 00185, Rome, Italy
Complexity Science Hub Vienna, Josefstädter Strasse 39, A-1080 Vienna, Austria